# Structure of papain-like protease from SARS-CoV-2 and its complexes with non-covalent inhibitors

Jerzy Osipiuk[1,2], Saara-Anne Azizi [3], Steve Dvorkin[4], Michael Endres[1,2], Robert Jedrzejczak[1,2], Krysten A. Jones [3], Soowon Kang[4], Rahul S. Kathayat [3], Youngchang Kim[1,2], Vladislav G. Lisnyak [3], Samantha L. Maki [3], Vlad Nicolaescu[4], Cooper A. Taylor[3], Christine Tesar[1,2], Yu-An Zhang[3], Zhiyao Zhou [3], Glenn Randall[4], Karolina Michalska[1,2], Scott A. Snyder [3✉], Bryan C. Dickinson [3✉] & Andrzej Joachimiak [1,2,5✉]

The pandemic caused by Severe Acute Respiratory Syndrome Coronavirus 2 (SARS-CoV-2) continues to expand. Papain-like protease (PLpro) is one of two SARS-CoV-2 proteases potentially targetable with antivirals. PLpro is an attractive target because it plays an essential role in cleavage and maturation of viral polyproteins, assembly of the replicase-transcriptase complex, and disruption of host responses. We report a substantive body of structural, biochemical, and virus replication studies that identify several inhibitors of the SARS-CoV-2 enzyme. We determined the high resolution structure of wild-type PLpro, the active site C111S mutant, and their complexes with inhibitors. This collection of structures details inhibitors recognition and interactions providing fundamental molecular and mechanistic insight into PLpro. All compounds inhibit the peptidase activity of PLpro in vitro, some block SARS-CoV-2 replication in cell culture assays. These findings will accelerate structure-based drug design efforts targeting PLpro to identify high-affinity inhibitors of clinical value.

[1] Center for Structural Genomics of Infectious Diseases, Consortium for Advanced Science and Engineering, University of Chicago, Chicago, IL, USA. [2] Structural Biology Center, X-ray Science Division, Argonne National Laboratory, Argonne, IL, USA. [3] Department of Chemistry, University of Chicago, Chicago, IL, USA. [4] Department of Microbiology, Ricketts Laboratory, University of Chicago, Chicago, IL, USA. [5] Department of Biochemistry and Molecular Biology, University of Chicago, Chicago, IL, USA. ✉email: sasnyder@uchicago.edu; dickinson@uchicago.edu; andrzejj@anl.gov

The Severe Acute Respiratory Syndrome Coronavirus 2 (SARS-CoV-2) is causing the COVID-19 pandemic. The virus belongs to the clade B of genus *Betacoronavirus*[1] and has large (+) sense ssRNA genome coding for 29 proteins. The four structural and 9–10 accessory proteins are translated from subgenomic RNAs produced from (−) sense ssRNA[2]. To reach the replication stage, the CoV-2 genomic (+) sense ssRNA is used as mRNA to ultimately produce 15 non-structural proteins (Nsps) from two large polyproteins, Pp1a (4405 amino acids) and Pp1ab (7096 amino acids)[3]. Pp1a is cleaved into the first ten Nsps (Nsp11 is just a seven residue peptide) and Pp1ab, which is made through a −1 ribosomal frame-shifting mechanism[4]. The resulting Pp1ab contains all 15 Nsps[5]. Therefore, proper polyprotein processing is essential for the release and maturation of the 15 Nsps and assembly into cytoplasmic, ER membrane-bound multicomponent replicase-transcriptase complex (RTC), which is responsible for directing the replication, transcription, and maturation of the viral genome and subgenomic mRNAs[6,7].

There are two distinctive cysteine proteases encoded by the SARS-CoV-2 genome that are essential to the virus proliferation cycle[6]: papain-like protease (PLpro, a domain within Nsp3, EC 3.4.22.46) and chymotrypsin-like main protease (3CLpro or Mpro, corresponding to Nsp5, EC 3.4.22.69). The main protease cuts 11 sites in Pp1a/Pp1ab with sequence consensus X-(L/F/M)-Q↓(G/A/S)-X[7,8] and PLpro cleaves three sites, with recognition sequence consensus "LXGG↓XX", but is as indispensable as Mpro because its activity extends far beyond polyproteins cleavage.

In SARS-CoV-2, Nsp3 contains 1945 residues (~212 kDa) 3. PLpro is a domain of Nsp3—a large multidomain protein that is an essential component of the RTC[7,8]. The enzyme is located in Nsp3 between the SARS unique domain (SUD/HVR) and a nucleic acid-binding domain (NAB). It is highly conserved and found in all coronaviruses[7], often in two copies, denoted as PL1pro and PL2pro[8,9]. This cysteine protease cleaves peptide bonds between Nsp1 and Nsp2 (LNGG↓AYTR), Nsp2 and Nsp3 (LKGG↓APTK), and Nsp3 and Nsp4 (LKGG↓KIVN) liberating three proteins: Nsp1, Nsp2, and Nsp3[10]. The LXGG motif found in Pp1a/Pp1ab corresponds to the P4–P1 substrate positions of cysteine proteases, and is essential for recognition and cleavage by PLpro[9,10]. Nsp1 is a 180 residue protein that interacts with 80 S ribosome and inhibits host translation[11,12]. Nsp2 is a 638 residue protein that was proposed to modulate host cell survival[13].

PLpro exhibits multiple proteolytic and other functions[14]. In addition to processing Pp1a/Pp1ab, it was shown in SARS- and MERS-CoVs to have deubiquitinating activity, efficiently disassembling mono-polyubiquitin, di-polyubiquitin, and branched-polyubiquitin chains. It also has deISG15ylating (interferon-induced gene 15) activities. Both ubiquitin and ISG15 protein carry the PLpro recognition motif at their C-termini[15,16] suggesting that removal of these modifications from host cells interferes with the host response to viral infection[9,17–20]. PLpro also inactivates TBK1, blocks NF-kappaB signaling, prevents translocation of IRF3 to the nucleus, inhibits the TLR7 signaling pathway, and induces Egr-1-dependent upregulation of TGF-β1[19,21]. Further illustrating the complex and diverse functions of the protein, in some reports, various PLpro roles are decoupled from its proteolytic activity[22,23]. Nevertheless, PLpro is a multifunctional protein having an essential role in processing of viral polyproteins, maturation, and assembly of the RTC, and it also may act on the host cell proteins by disrupting host viral response machinery to facilitate viral proliferation and replication. Due to the centrality of PLpro to viral replication, it is an excellent candidate for therapeutic targeting.

Ongoing efforts to identify antivirals for SARS-CoV-2 to date have focused mainly on three Nsp proteins identified as the key drug targets from previous SARS-CoV and MERS-CoV studies: Nsp3 PLpro, Nsp5 Mpro, and Nsp12 RNA-dependent RNA polymerase. Here we discuss the case for targeting SARS-CoV-2 Nsp3 domain, PLpro. The enzyme is conserved in SARS-CoV, MERS-CoV, Swine Acute Diarrhea Syndrome (SADS, PLP2 enzyme) coronaviruses (Supplementary Fig. 1), and other viruses including Murine Hepatitis Virus, Avian Infectious Bronchitis Virus, and Transmissible Gastroenteritis Virus (TGEV); fortuitously, it has low sequence similarity to human enzymes. The sequence, structure, and functional conservation of PLpro suggests that therapeutics targeting SARS-CoV-2 PLpro may also be effective against related viruses with PLpro. In the past, this enzyme was structurally well characterized and currently there are over 40 structures of viral PLpro proteases in the Protein Data Bank (PDB)[24], mainly from SARS-CoV, that can aid structure-based drug discovery. In fact, the past 15 years of studies with PLpro have led to the identification of a number of inhibitors that are specific for SARS-CoV PLpro, but do not inhibit the MERS-CoV enzyme[25–27]. Unfortunately, these efforts have failed, thus far, to produce antivirals that can be useful for treatment of SARS-CoV and SARS-2 infections in humans.

Here we report seven crystal structures, including the high-resolution structure of wild-type PLpro, the active site cysteine mutant, and their complexes with inhibitors of SARS-CoV-2 PLpro, determined at 1.60–2.50 Å. These data reveal the structural basis of the enzyme with fine molecular details, and illustrate specific ligand recognition and interactions. The presented compounds inhibit PLpro peptidase activity in vitro, and most importantly, several of the identified inhibitors also block SARS-CoV-2 replication in cell culture. Collectively, these findings provide critical insights for further structure-based drug design efforts against PLpro to enable the design of even higher affinity inhibitors and, ultimately, human therapeutics.

## Results and discussion

The SARS-CoV-2 PLpro sequence is 83% identical and 90% similar to SARS-CoV and 31% identical and 49% similar to MERS-CoV and even more distant to SADS PLP2 (Supplementary Fig. 1). SADS is an alphacoronavirus and contains two papain-like proteases PLP1 and PLP2 and only structure of PLP2 (PDB id: 6L5T) is available. Between SARS-CoV and MERS-CoV many substitutions are quite conservative, but there are some that may have significant impact on protein stability, dynamics, ligand binding, and catalytic properties. Examples include Thr75 (Leu/Val), Pro129 (Ala/Ile), Tyr172 (His/Thr), Lys200 (Thr/Gln), Lys274 (Thr/Val), and Cys284 (Arg/Arg), with equivalent residues shown in parentheses for SARS-CoV/MERS-CoV PLpro, respectively.

SARS-CoV-2 PLpro is a slightly basic, 315 residue protein with high content of cysteine residues (3.5%). In addition to catalytic Cys111, there are four cysteine residues coordinating important structural zinc ion and other six distributed throughout the protein structure. Similar to Mpro[28,29], the active site cysteine seems much more reactive as evident by structures of covalent adducts reported in the PDB (PDB id: 6WX4 and 6WUU). SARS-CoV-2 Mpro-active cysteine has been shown to have different level of oxidation in crystals (PDB id: 6XKF, 6XKH[28]). The potential sensitivity of PLpro Cys111 to oxidation presented a challenge for structure determination as the wild-type protein exhibited rather poor crystallization properties. The PLpro active site contains a canonical cysteine protease catalytic triad (Cys111, His272, and Asp286)[18], while Mpro has catalytic dyad (Cys145 and His41)[29], which may account for somewhat dissimilar chemical properties of the two enzymes. PLpro may have catalytic properties more common with other cysteine proteases, with the generally accepted thiolate form of Cys111 acting as a nucleophile and Asp286 promoting deprotonation of His272, which serves as a base. In agreement with previous suggestions[30] our structures

point to a potential oxyanion hole provided by main chain amides of Cys111/Tyr112. The amides would be at the distance ~3 Å from oxygen atom.

**Structure determination and structural comparisons**. We report here seven structures of PLpro from SARS-CoV-2, including wild-type apo-protein structure determined at 100 K and refined to 1.79 Å (PDB id: 6WZU), the apo-PLpro active site C111S mutant under cryogenic conditions 100 K at 1.60 Å (PDB id: 6WRH) and at 293 K at 2.50 Å (PDB id: 6XG3). The first structure (wild-type) was solved by molecular replacement using SARS PLpro model as a template. The subsequent structures were phased with the refined wild-type model. Structures were refined as described in Methods and data and refinement statistics is shown in Supplementary Table 1. For high resolution structures, all residues are visible in the electron density maps and for the 293 K 2.50 Å mutant structure two N-terminal and one C-terminal residues are missing. Of significance, the electron density map for residues around the active site is excellent. These structures are virtually identical with the high resolution wild-type and C111S mutant structures showing RMSD 0.10 Å, while 293 and 100 K mutant structures show RMSD 0.27 Å. They differ the most in the zinc-binding region and in the Gly266–Gly271 loop containing Tyr268 and Gln269. In the high resolution structures, in addition to structural zinc ion, there are two chloride and two phosphate ions bound. In the highest resolution structure we modeled 381 water molecules.

Structures of Nsp3 PLpro were reported previously for SARS-CoV, MERS-CoV, and other viruses[31]. The SARS-CoV-2 PLpro structure has "thumb–palm–fingers" architecture described before (Fig. 1A). This arrangement is similar to the ubiquitin specific proteases (USPs, Fig. 1B), one of the five distinct deubiquitinating enzyme (DUB) families, despite low sequence identities (~10%)[9,31]. Briefly, the protein has two distinct domains: the small N-terminal ubiquitin-like (Ubl) domain and the "thumb–palm–fingers" catalytic domain (Fig. 1). The Ubl domain consists of residues 1–60 with five β-strands, one α-helix, and one $3_{10}$-helix. In SARS-CoV-2 PLpro, a chloride ion binds to a small loop formed by residues Thr9–Ile14 at the interface with the catalytic domain. In different structures, Ubl shows some conformational flexibility, though the specific function of this domain is not well understood.

The larger catalytic domain is an extended right-hand scaffold with three characteristic subdomains. A thumb is comprised of six α-helices and a small β-hairpin. The fingers subdomain is the most complex; it is made of six β-strands and two α-helices and includes a zinc binding site. This structural zinc ion is coordinated by four cysteine residues located on two loops (Cys189, 192, 224, and 226) of two β-hairpins. Zinc binding is essential for structural integrity and protease activity[32], but the conformation of this region varies most between different PLpro structures. The palm subdomain is comprised of six β-strands (Fig. 1) with the catalytic residues Cys111, His272, and Asp286 located at the interface between the thumb and palm subdomains (Fig. 1). An important mobile β-turn/loop (Gly266–Gly271) is adjacent to the active site that closes upon substrate and/or inhibitor binding. In the high resolution structure of PLpro C111S mutant, there is a phosphate ion bound to the active site at the N-terminus of helix α4 (contributing Cys111) that is coordinated by Trp106, Asn109, and His272. This site provides a good environment to stabilize C-terminal carboxylate group of the peptide cleavage product. In this structure, there is another phosphate ion bound to a thumb subdomain and is coordinated by His73 and His170. This subdomain also binds a second chloride ion near Arg140. In the structures with inhibitors there are additional zinc and chloride ions bound, including one in the active site that is coordinated by the active site Cys111.

We compared our structures with the high resolution crystal structures of SARS-CoV and MERS-CoV PLpro. The SARS PLpro Cys112Ser mutant in complex with ubiquitin (PDB id: 4M0W) shows RMSD 0.53 Å with our highest resolution structure of SARS-CoV-2 PLpro C111S mutant (PDB id: 6WRH). The largest differences are observed in zinc binding region, consistent with this region being the most flexible in the PLpro structures. A comparison of MERS PLpro high resolution structure (PDB id: 4RNA) with SARS-CoV-2 PLpro C111S mutant shows much bigger differences (RMSD 1.82 Å), with the largest structural shifts occurring again in the structural zinc-binding region and in the N-terminal Ubl domain, but also in the palm subdomain (Supplementary Fig. 5). The PLpro core shows analogous differences with RMSD 1.72 Å (PDB id: 5KO3). In compared structures, as expected, side chains of many surface residues show different conformations. Nevertheless, the arrangement of the catalytic site is very similar in SARS-CoV-2 and SARS-CoV, suggesting that at least some inhibitors may display cross activity between these proteases. The MERS-CoV PLpro active site region differs quite significantly from SARS PLpro enzymes and at least some SARS-specific inhibitors may not cross react.

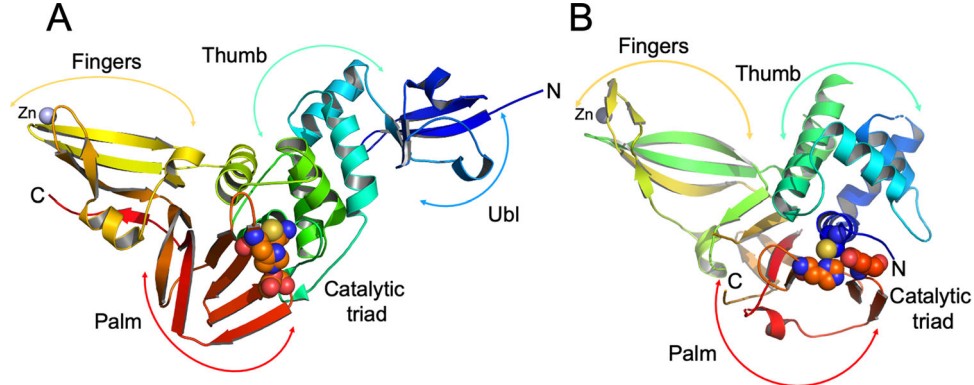

**Fig. 1 Structure of PLpro from SARS-CoV-2 and comparison with human USP12. A** PLpro model showing secondary structure and "thumb-palm-fingers" architecture, with domains and subdomains labeled and active site residues Cys111/His272/Asp286 represented as spheres, zinc ion is in blue. **B** Model of human USP12 (PDB id: 5K1B[48]) showing similar fold and domain architecture with active site residues Cys48/His317/Asp333 represented as spheres and zinc ion in blue.

**Enzyme activity assays of synthetic compounds.** Several naphthalene-based compounds were synthesized (Supplementary Fig. 2) and tested for inhibition of SARS-CoV-2 PLpro activity (Supplementary Fig. 3). One of these compounds (1/GRL0617) was identified previously as a specific SARS-CoV PLpro inhibitor, and showed good potency and low cytotoxicity in SARS-CoV-infected Vero E6 cells[33]. We present in this manuscript results of biochemical, whole cell, and high resolution crystallographic studies of seven compounds, six possessing the methyl-$N$-[(1$R$)-1-naphthalen-1-ylethyl]benzamide scaffold and one being a simplified analog of our own design. All these compounds inhibit SARS-CoV-2 PLpro protease and they are designated as follows: **1** is 5-amino-2-methyl-$N$-[(1$R$)-1-naphthalen-1-ylethyl]benzamide (GRL0617), **2** is 5-carbamylurea-2-methyl-$N$-[(1$R$)-1-naphthalen-1-ylethyl]benzamide, **3** is 5-acrylamide-2-methyl-$N$-[(1$R$)-1-naphthalen-1-ylethyl]benzamide, **4** is 3-amino-$N$-(naphthalene-1-yl)-5-trifluoromethyl)benzamide, **5** is 5-(butylcarbamoylamino)-2-methyl-$N$-[(1$R$)-1-naphthalen-1-ylethyl]benzamide, **6** is 5-(((4-nitrophenoxy)carbonyl)amino)-2-methyl-$N$-[(1$R$)-1-naphthalen-1-ylethyl]benzamide, and **7** is 5-pentanoylamino-2-methyl-$N$-[(1$R$)-1-naphthalen-1-ylethyl]benzamide (Fig. 2 and Supplementary Fig. 2A).

We have developed an in vitro biochemical assay for PLpro using expressed protein and a pro-fluorescent peptide substrate, CV-2, designed based on the LKGG recognition motif of PLpro found in SARS-CoV-2 polyproteins (Supplementary Figs. 2B and 3). CV-2 generates fluorescent signal in response to the protease activity of PLpro, and critically, is unresponsive in the C111S variant (Supplementary Fig. 3). In this assay all seven compounds act as non-covalent inhibitors of the enzyme. Given that compound 1/GRL0617 is known to inhibit SARS-CoV PLpro with an $IC_{50}$ value of 0.6 μM[33], we expected it would likely inhibit the SARS-CoV-2 enzyme and indeed it does so, with an $IC_{50}$ value of 2.3 μM in our assay conditions (Fig. 2). Compounds **2**, **3**, **5**, **6**, and **7** are further amine-functionalized derivatives of **1**/GRL0617, while **4** is a simplified variant of **1** without a chirality center (Supplementary Fig. 2A); despite their structural differences, we found that all inhibit PLpro to varying degrees ($IC_{50} = 5.1$–32.8 μM, Fig. 2). Given this suite of molecules that function as PLpro inhibitors in vitro, we next sought to test whether these molecules are also capable of inhibiting viral replication in live cells.

**Whole cell virus replication assays.** We next performed SARS-CoV-2 virus replication assays using Vero E6 cells and measuring

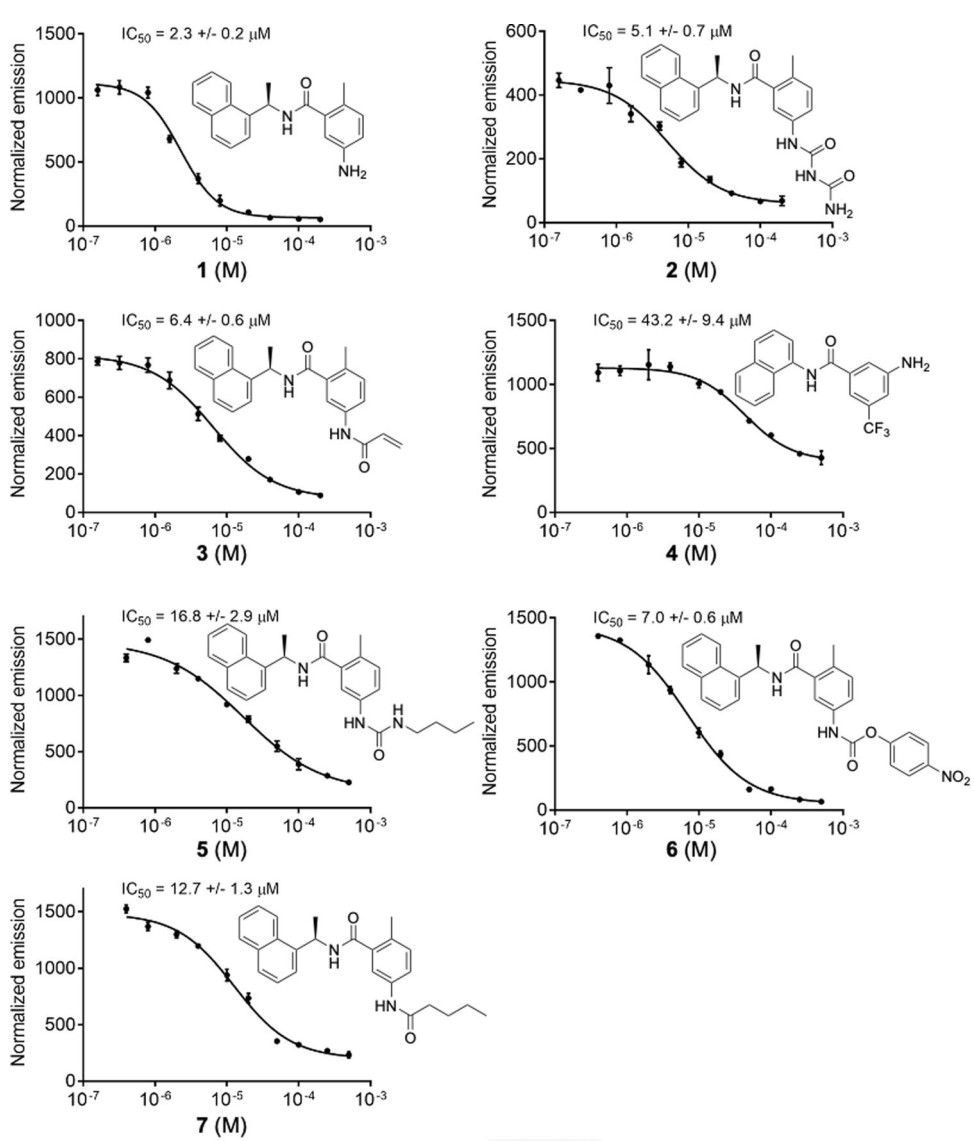

**Fig. 2 Biochemical activity assays for compounds 1–7.** Activity assays were performed using substrate shown in Supplementary Fig. 3. Error bars represent the standard error of the mean for $n = 3$ replicates.

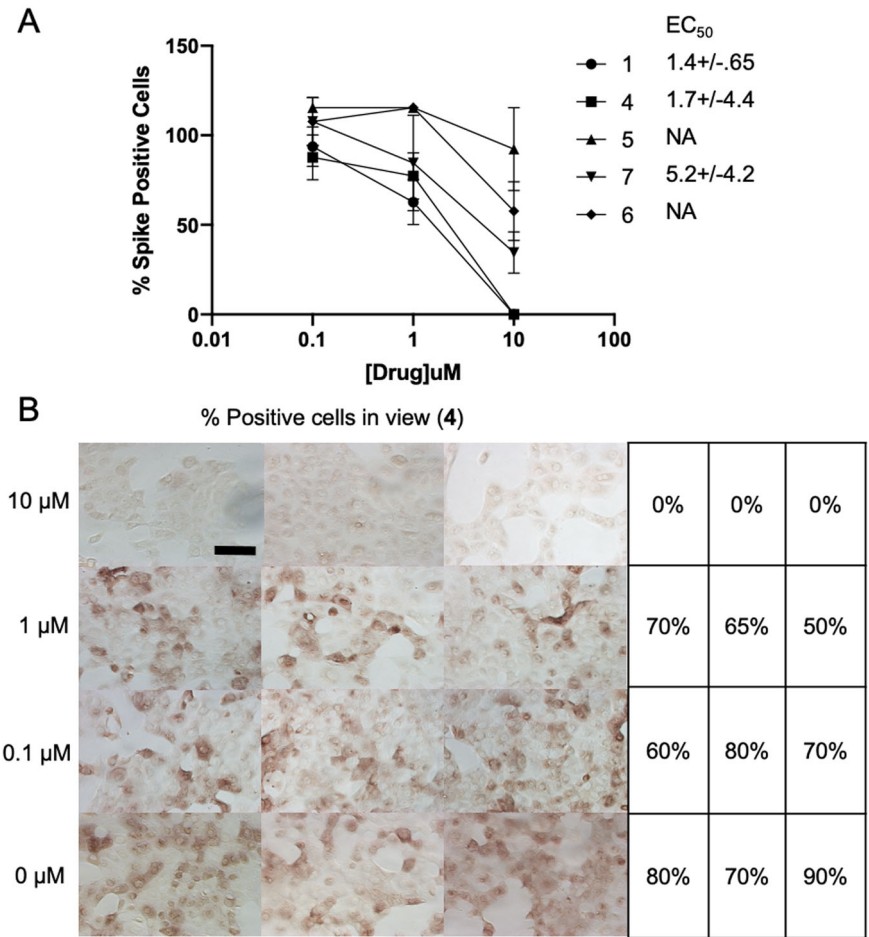

**Fig. 3 Virus inhibition in whole cell assay. A** Virus replication activity assays for compounds **1**, **4**, **5**, **6**, and **7**. Data are mean % percent spike positive cells relative to DMSO treated cells $+/-$ SEM of three biological replicates. **B** whole cell assay for compound **4**. Percent Spike positive cells, $n = 100$. Scale bar is 100 μM.

SARS-CoV-2 replication. Surprisingly, not all compounds functioned in this assay, and their relative abilities to inhibit viral replication did not necessarily correlate directly with in vitro inhibition parameters toward PLpro. Compounds **1**, **4**, **5**, **6**, and **7** all proved capable of affecting the viability of cells and inhibiting virus replication. For compounds **1**, **4**, and **7** their $EC_{50}$ values range from 1.4 to 5.2 μM (Fig. 3A, B and Supplementary Fig. 4). For example, PLpro inhibition by compound **1**/GRL0617 is 14-times better than **4**, but inhibition of virus replication is only two times higher. Moreover, compounds **2** and **3** are quite good PLpro inhibitors ($IC_{50}$ values of 5.1 and 6.4 μM, respectively), but failed in the viral replication assay. Compound **5** was the weakest inhibitor in vitro ($IC_{50}$ values of 32.8 μM), but was one of the best performers in the live viral replication assay ($EC_{50} = 2.5$ μM).

Differences in cell permeability and solubility could account for the disconnects between the in vitro biochemical assay data and viral replication data, but given the high degree of structural similarity between these molecules, these data indicate that further optimization is possible, especially in the case of compound **5**, which is a relatively weak binder but solid inhibitor of the virus. More broadly, all of the compounds are promising and may need only small modification(s) in order to serve as preclinical lead compounds. To enable structure-based improvements of these molecules, we next aimed to get ligand-bound structures of as many of these lead compounds as possible. Based on these results, we believe it is critical to combine the in vitro biochemical assays to triage compounds with live viral replication assays.

**Crystal structures with bound compounds 1, 2, and 3**. We were able to determine four crystal structures of SARS-CoV-2 PLpro with non-covalent inhibitors **1**, **2**, and **3** (Supplementary Table 1, Supplementary Fig. 6, Supplementary Methods, and Supplementary Data 1) three using C111S mutant and one wild-type enzyme (PDB ids: 7JIR, 7JIT, 7JIV, and 7JIW). The structure with compound **3** was solved in both wild-type and mutant forms. The electron density for the ligand, protein, solvent, and bound acetate ion is excellent (Fig. 4). All three compounds bind to the same site as observed previously for GRL0617 in complex with SARS-CoV PLpro (PDB id: 3E9S)[33] and now determined for **1** in complex with SARS-CoV-2 PLpro (Fig. 4A and Supplementary Fig. 6A). The structure of **2** bound to the PLpro C111S mutant was determined at 1.95 Å (PDB id: 7JIT), the highest resolution for all complexes to date, and this structure will be used here as a reference (Fig. 4B, Supplementary Fig. 6B, and Supplementary Fig. 7A). Compound **2** binds to the groove on the surface of PLpro near the active site, ~8 Å apart from Cys/Ser111, overlapping with S4/S3 protein subsites (corresponding the substrate positions P4/P3), that are critical for recognition of the leucine residue in the LXGG motif (Fig. 4E). The ubiquitin peptide-binding site is a solvent-exposed groove: wide at S4 site, solvent exposed at P3 site, and very narrow at P2 and P1 sites (Fig. 4E). Because of the high resolution achieved, we observe extensive interactions between compound **2** and the protein involving direct as well as water-mediated hydrogen bonds and van der Waals contacts, noting some additional interactions provided by

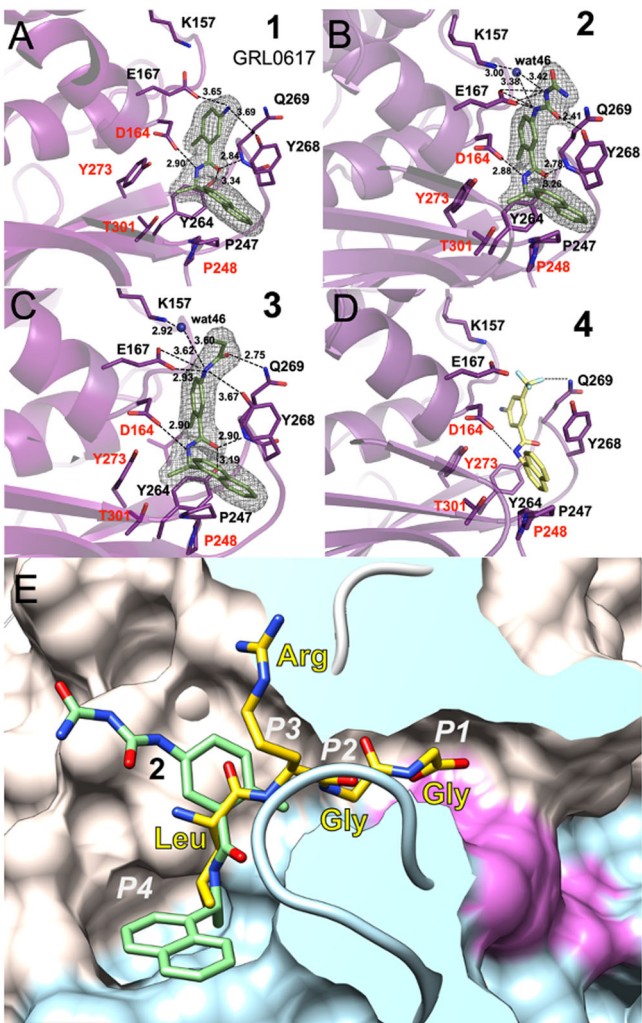

**Fig. 4 Ligands binding to SARS-CoV-2 PLpro. A** Compound **1** binding to PLpro. **B** Compound **2** binding to PLpro. **C** Compound **3** binding to PLpro. **D** Model of compound **4** (yellow sticks) binding to PLpro. Ligands are shown as green sticks and PLpro is in magenta. Dashed lines show hydrogen bonds, water molecules are shown as blue spheres. In **A**–**C** the $2F_o - mF_c$ electron density maps are shown as a grey mesh, contoured at 1.2 $\sigma$. **E** Compound **2** (green sticks) binds to a groove on the surface of PLpro protein (surface of palm subdomain is in white and thumb subdomain is in light blue) with the active site catalytic triad surface is shown in red in the end of a slender tunnel. Peptide LRGG from ubiquitin structure in complex with SARS PLpro (PDB id: 4MOW) is shown in yellow and peptide positions corresponding P1–P4 sites are marked in white.

the carbamylurea moiety in **2**. As compared with unliganded protein, the main chain and side chains of several residues significantly adjust to accommodate the ligand (Arg166, Glu167, Tyr268, Gln269, Fig. 4B and Supplementary Fig. 7A). Direct hydrogen bonds are found between Glu167 and Tyr268 and two nitrogen and oxygen atoms of the carbamylurea moiety, respectively (Fig. 4B). As compared with **1**/GRL0617 compound **2** makes all the same interactions and provides additional four hydrogen bonds. Intriguingly, however, the IC$_{50}$ value for **1** is ~2 times lower than for **2**, despite **2** making more interactions with the protein. Asp164 hydrogen bonds with another amino group in the linker between two aromatic rings and main-chain amino group of Gln269 hydrogen bonds to oxygen atom of that linker.

Lys157 makes water-mediated hydrogen bond to the carbamylurea moiety. This water molecule also coordinates Glu167. Interestingly, this residue (equivalent of Glu168 in SARS PLpro) seems to play an important role in Ub1 core recognition, and mutations can cause a significant loss of DUB activity.

The aromatic rings of **2** make several hydrophobic interactions, specifically with Pro248, Tyr268, aliphatic regions of the side chain of Gln269, and Asp164. Curiously, there is also an acetate ion packing in between **2** and protein residues. This ion is coordinated by Arg166 and Glu167. The ligand binding site offers a number of opportunities to improve ligand affinity (water and acetate binding site) and potential linking to the active site for covalent attachment, many of which were not explicitly indicated by previous work with SARS.

The structures of inhibitor **3** were determined in both forms: wild-type at 2.30 Å (PDB id: 7JIW) and C111S mutant at 2.05 Å (PDB id: 7JIV) (Fig. 4C and Supplementary Fig. 6C, D). The structures of protein, the pose, and overall interactions of the inhibitor are the same in both variants. Compound **3** is a derivative of **1**, in which its amino moiety is functionalized to become an acrylamide. Structural comparison shows that only some interactions are preserved—all hydrophobic interactions and one hydrogen bond. The acrylamide moiety in **3** provides two additional hydrogen bonds as compared to **1**. Thus far, we have not been able to grow crystals of SARS-CoV-2 PLpro with compound **4**. However, we were able to model the interaction of **4** with the protein (Fig. 4D), where it appears that the trifluoromethyl moiety is able to interact with the amide group of Gln269. Given that several other analogs of **4** are inactive in biochemical studies, this interaction might be significant.

The inhibitory effect of compounds **1**, **2**, **3**, and **4** can be easily rationalized as they anchor to the site. Although somewhat away from the catalytic triad, their binding still interferes with the recognition of peptide motif LXGG. Comparison with the high resolution structure of PLpro with ubiquitin (PDB id: 4M0W) shows that the inhibitor linker region connecting naphthalene and benzene rings overlaps with the leucine residue of ubiquitin C-terminal sequence bound to the S4 subsite (Fig. 4E and Supplementary Fig. 7B). The S4 site is where specificity of the LXGG peptide is determined as it recognizes leucine side chain by fitting it into hydrophobic pocket formed by Pro248, Tyr264, Tyr272, and Thr301. These residues are conserved in SARS-CoV and SARS-CoV-2 PLpro, but only Tyr272 and Thr301 are conserved in SARS-CoV and MERS-CoV PLpro (Supplementary Fig. 1). Therefore, ligands that bind to hydrophobic S4 site and make hydrophilic interactions with PLpro surface residues may be good candidates for inhibitors of the PLpro enzyme. The S3 site can accept any residue because it is solvent exposed, but would prefer hydrophilic side chain (Arg, Lys, and Asn). The peptide then follows the path to the active site that becomes narrower and can accept only two glycine residues in P1 and P2. The P1′ again is on the protein surface and can accept any residue. The interesting requirement is that the peptide binding to S1–S4 sites must be in extended/linear conformation, placing noteworthy constrains on designing inhibitors targeting the active site.

In summary, in this report, we have presented a substantial high resolution structural, biochemical, and virus replication studies of PLpro cysteine protease from SARS-CoV-2 and describe seven compounds that inhibit enzyme in in vitro biochemical assay based on the cleavage of the LKGG recognition peptide, a subset of these inhibit virus replication. We have determined apo-structures of wild-type enzyme and inactive mutant in which single sulfur atom was replaced by oxygen (C111S). The apo and mutant structures were determined at high

resolution 1.79 and 1.60 Å providing detailed and accurate three-dimensional models of the enzyme. The mutant structures were determined under both 100 and 293 K temperatures providing information about protein flexibility. All four apo-PLpro structures showed significant structural conservation, including catalytic triad and ordered solvent molecules. The protein surface provides rich chemical environment and is capable of binding a variety of ions including conserved structural zinc ion, few non-structural zinc ions, several phosphate, chloride, and acetate anions. The structures of complexes with inhibitors **1**, **2**, and **3** were determined at 1.95–2.30 Å, including compound **3** in both wild-type and mutant forms. All three ligands bind to the same site in the enzyme located 8–10 Å away from the catalytic cysteine and it is expected that all seven synthetic compounds bind in a very similar manner. Based on this assumption we have modelled pose of compound **4** in the structure. Considerable conformational adjustments are observed for the side chains of residues involved in ligand binding. These inhibitors bind to protease S4/S3 sites. The S4 site is where specificity of the LXGG sequence recognition motif is determined as it recognizes leucine side chain by fitting it into hydrophobic pocket. This site is only partly conserved between SARS-CoV and MERS-CoV enzymes explaining lack of cross reactivity of compound **1** reported previously. Binding inhibitors to this site would block peptide recognition. The PLpro peptide binding site narrows significantly as it approaches catalytic triad explaining why only glycine residues are accepted at the C-terminus of the recognition motif. These compounds would not only prevent virus polyproteins processing but also cleavage of host proteins modifications with ubiquitin and ISG15, therefore inhibit several PLpro functions. Five out of seven inhibitors of PLpro block virus replication in whole cell assay. Interestingly, their relative abilities to inhibit viral replication do not directly correlate with in vitro inhibition of PLpro, suggesting that other factors are important. Nevertheless our studies showed potential S4/S3 site binders to serve as scaffolds for effective inhibitors of SARS-CoV-2 coronavirus. Our collection of structures provides fundamental molecular and mechanistic insight into PLpro structure, and it illustrates details ligand recognition and interactions. These collated findings will accelerate further structure-based drug design efforts targeting PLpro, with the ultimate goal of identifying high-affinity inhibitors of clinical value for SARS-CoV-2.

## Methods

**Gene cloning, protein expression and purification of WT and C111S mutant of PLpro**. The gene cloning, protein expression and purification were performed using protocols published previously[34]. Briefly, the Nsp3 DNA sequence corresponding to PLpro protease SARS-CoV-2 was optimized for *E. coli* expression using the OptimumGene codon optimization algorithm followed by manual editing and then cloned directly into pMCSG53 vector (Twist Bioscience) (Supplementary Table 2). The plasmids were transformed into the *E. coli* BL21(DE3)-Gold strain (Stratagene). *E. coli* cells harboring plasmids for SARS CoV-2 PLpro WT and C111S mutant expression were cultured in LB medium supplemented with ampicillin (150 μg/ml).

Bacterial cells were harvested by centrifugation at $7000 \times g$ and cell pellets were resuspended in a 12.5 ml lysis buffer (500 mM NaCl, 5% (v/v) glycerol, 50 mM HEPES pH 8.0, 20 mM imidazole pH8.0, 1 mM TCEP, 1 μM ZnCl$_2$) per liter culture and sonicated at 120 W for 5 min (4 s ON, 20 s OFF). The cellular debris was removed by centrifugation at $30,000 \times g$ for 90 min at 4 °C. The supernatant was mixed with 3 ml of Ni$^{2+}$ Sepharose (GE Healthcare Life Sciences) which had been equilibrated with lysis buffer supplemented to 50 mM imidazole pH 8.0, and the suspension was applied on Flex-Column (420400-2510) connected to Vac-Man vacuum manifold (Promega). Unbound proteins were washed out via controlled suction with 160 ml of lysis buffer (with 50 mM imidazole pH 8.0). Bound proteins were eluted with 15 ml of lysis buffer supplemented to 500 mM imidazole pH 8.0, followed by Tobacco Etch Virus (TEV) protease treatment at 1:25 protease:protein ratio. The solutions were left at 4 °C overnight. The proteins were run separately on a Superdex 75 column equilibrated in lysis buffer. Fractions containing cut protein were collected and applied on Flex-Columns with 3 ml of Ni$^{2+}$ Sepharose which had been equilibrated with lysis buffer. The flow through and a 7 ml lysis buffer

rinse were collected. Lysis buffer was replaced using 30 kDa MWCO filters (Amicon-Millipore) via 10× concentration/dilution repeated three times to crystallization buffer (150 mM NaCl, 20 mM HEPES pH 7.5, 1 μM ZnCl$_2$, 4 mM TCEP). Purification was repeated for PLpro WT and C111S mutant proteins for co-crystallization with inhibitors, following the same protocol except that 10 mM β-mercaptoethanol was used instead of TCEP in all purification buffers, and 10 mM DTT was used instead of TCEP in the crystallization buffer. The final concentrations of WT PLpro was 25 mg/ml and C111S mutant was 30 mg/ml.

**Fluorescence-based biochemical assays**. Dose response assays were performed in 96-well plate format in triplicate at 25 °C. Wells containing varying concentrations of PLpro enzyme (0–1 μM) in Tris-HCl pH 7.3, 1 mM EDTA were mixed with LKGG-AMC probe substrate (40 μM) and measured continuously for fluorescence emission intensity (excitation λ: 364 nm; emission λ: 440 nm) on a Synergy Neo2 Hybrid. PLpro-WT and PLpro-C111S activities on LKGG-AMC were assayed as above with 1 μM enzyme and 40 μM LKGG-AMC substrate. PLpro-WT activity on LKGG-AMC probe in the presence and absence of EDTA was assayed as above with 1 μM enzyme and 40 μM LKGG-AMC substrate.

**PLpro inhibition assay**. Inhibition assays were performed in a 96-well plate format in triplicate at 25 °C. Reactions containing varying concentrations of inhibitor (0–500 μM) and PLpro enzyme (0.3 μM) in Tris-HCl pH 7.3, 1 mM EDTA were incubated for approximately 5 min. Reactions were then initiated with LKGG-AMC probe substrate (40 μM), shaken linearly for 5 s, and then measured continuously for fluorescence emission intensity (excitation λ: 364 nm; emission λ: 440 nm) on a Synergy Neo2 Hybrid. Data were fit using nonlinear regression (dose-response inhibition, variable slope) analysis in GraphPad Prism 7.0.

**PLpro crystallization**. The sitting-drop vapor-diffusion method was used with the help of the Mosquito liquid dispenser (TTP LabTech) in 96-well CrystalQuick plates (Greiner Bio-One). Crystallizations were performed with the protein-to-matrix ratio of 1:1. MCSG1, MCSG2, MCSG3, and MCSG4 (Anatrace) screens were used for protein crystallization at 4 and 16 °C. The crystals of PLpro-C111S mutant protein (bipyramidal crystals up to 0.2 mm, 1–3 days of incubation at 4 °C, P3$_2$21 space group) were obtained from MCSG2 screen, reagent formulation #4 (0.1 M acetate buffer pH 4.5, 0.8 M NaH$_2$PO$_4$/1.2 M K$_2$HPO$_4$). The mutant crystals were used for seeding WT protein crystallization droplets to obtain crystals with significantly improved diffraction. For co-crystallization with inhibitors, proteins (15 mg/ml) were mixed with inhibitors at 10× protein concentration for a final inhibitor concentration of 4 mM, incubated on ice for 2.5 h, and spun down to remove precipitation. Crystals (I4$_1$22 space group) formed at 4 °C with a protein-to-matrix ratio of 2:1 in hanging drops in 0.1 M MES pH 6.0, 50 mM zinc acetate, 10% PEG 8000. Crystals selected for data collection were washed in the crystallization buffer supplemented with either 25% glycerol (apo-protein crystals) or 25% ethylene glycol (protein-inhibitor crystals) and flash-cooled in liquid nitrogen.

**Data collection, structure determination, and refinement**. Single-wavelength x-ray diffraction data were collected at 100 K temperature at the 19-ID beamline of the Structural Biology Center at the Advanced Photon Source at Argonne National Laboratory using the program SBCcollect. The diffraction images were recorded from all crystal forms on the PILATUS3 X 6M detector at 12.662 keV energy (0.9792 Å wavelength) using 0.3° rotation and 0.3 s exposure. The intensities were integrated and scaled with the HKL3000 suite[35]. Intensities were converted to structure factor amplitudes in the truncate program[36,37] from the CCP4 package[38]. The structures were determined by molecular replacement using HKL3000 suite incorporating the following program MOLREP[39–41]. The initial solutions were refined, both rigid-body refinement and regular restrained refinement by REFMAC program[42] as a part of HKL3000. The coordinates of SARS coronavirus PLpro (PDB id: 5Y3Q) were used as the starting model for the first wild-type protein structure solution. Several rounds of manual adjustments of structure models using COOT[43] and refinements with REFMAC program[42] from CCP4 suite[38] were done. The models including the ligands were manually adjusted using COOT and then iteratively refined using COOT and REFMAC. The stereochemistry of the structure was validated with PHENIX suite[44] incorporating MOLPROBITY[45] and PRO-CHECK[46] tools. Throughout the refinement, the same 5% of reflections were kept out from the refinement. The stereochemistry of the structure was checked with the Ramachandran plot and validated with the PDB validation server. No Ramachandran plot outliers were observed in the structures with one exception, His47 in the structure of WT PLpro protease complexed with compound **3**. Depending on structure quality, 93.5–97.5% of PLpro residues were in Ramachandran plot favored regions. A summary of data collection and refinement statistics is given in Supplementary Table 1.

**NMR spectra**. $^1$H NMR and $^{13}$C NMR spectra were collected at 25 °C on 400 MHz Bruker DRX400 at the Department of Chemistry NMR Facility at the University of Chicago. $^1$H-NMR chemical shifts are reported in parts per million (ppm) relative to the peak of residual proton signals from (CDCl$_3$ 7.26 ppm or DMSO-d$_6$ 2.50 ppm). Multiplicities are given as: s (singlet), d (doublet), t (triplet), q (quartet), dd (doublet of doublets), p (pentet), m (multiplet), and br[47]. $^{13}$C-NMR chemical

shifts are reported in parts per million (ppm) relative to the peak of residual proton signals from (CDCl$_3$ 77.16 ppm). Analysis of NMR was done in MestReNova (version 14.1.2–25024). High resolution mass was obtained from Agilent 6224 TOF High Resolution Accurate Mass Spectrometer (HRA-MS) using combination of APCI and ESI at the Department of Chemistry Mass Spectrometry Facility at the University of Chicago. Low resolution mass spectral analyses and liquid chromatography analysis were carried out on an Advion Expression-L mass spectrometer (Ithaca, NY) coupled with an Agilent 1220 Infinity LC System (Santa Clara, CA).

**Cells and virus.** Vero E6 cells (ATCC) were infected under biosafety level 3 conditions with SARS-CoV-2 (nCoV/Washington/1/2020, kindly provided by the National Biocontainment Laboratory, Galveston, TX).

**Immunostaining against Spike protein.** Vero E6 cells were treated with the indicated concentration of inhibitor in DMSO for 2 h, then infected with SARS-CoV-2 at a multiplicity of 0.1 for 24 h in the presence of inhibitor. Infected cells were fixed with 10% NBF in 96-well plate. After fixation, 10% NBF was removed, and cells were washed with PBS, followed by washing with PBS-T (0.1% Tween 20 in PBS), and then blocked for 30 min with PBS containing 1% BSA plus 0.125% Saponin at 23 °C. After blocking, endogenous peroxidases were quenched by 3% hydrogen peroxide for 5 min. Then, cells were washed with PBS and PBS-T and incubated with a monoclonal mouse-anti-SARS-CoV-2 spike antibody ([1A9] Catalog number GTX632604, GeneTex, 1:3000 dilution) in PBS containing 1% BSA overnight at 4 °C. Primary antibody was washed with PBS and PBS-T and then cells were incubated in secondary antibody (ImmPRESS Horse Anti-Mouse IgG Polymer Reagent, Peroxidase, catalog number MP-7402-50; Vector Laboratories, 1:3 dilution) for 60 min at 23 °C. After washing with PBS for 10 min, color development was achieved by applying diaminobenzidine tetrahydrochloride (DAB) solution (Metal Enhanced DAB Substrate Kit; ThermoFisher Scientific) for 30 min and observed by light microscopy for the percentage of cells that were Spike positive.

**Reporting summary.** Further information on research design is available in the Nature Research Reporting Summary linked to this article.

## Data availability
The structural datasets generated during the current study are available in the Protein Data Bank repository (https://www.rcsb.org/) under accession codes: 6WZU, 6WRH, 6XG3, 7JIR, 7JIT, 7JIV, and 7JIW. Diffraction images are available on server in Dr. W. Minor laboratory https://proteindiffraction.org. A Source Data file is provided with this manuscript. Plasmid for expression PLpro (NR-52897 Vector pMCSG53 Containing the SARS-Related Coronavirus 2, Wuhan-Hu-1 Papain-Like Protease) is available in the NIH the BEI Resources Repository (https://www.niaid.nih.gov/research/bei-resources-repository). All other data generated during the current study including the raw kinetic and biophysical data are available upon request. Source data are provided with this paper.

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

## Acknowledgements
We thank the members of the SBC at Argonne National Laboratory, especially Darren Sherrell and Alex Lavens for their help with setting beamline and data collection at beamline 19-ID. Funding for this project was provided in part by federal funds from the National Institute of Allergy and Infectious Diseases, National Institutes of Health, Department of Health and Human Services, under Contract HHSN272201700060C (to A.J.) and by the DOE Office of Science through the National Virtual Biotechnology Laboratory, a consortium of DOE national laboratories focused on response to COVID-19, with funding provided by the Coronavirus CARES Act (to A.J.). The use of SBC beamlines at the Advanced Photon Source is supported by the U.S. Department of Energy (DOE) Office of Science and operated for the DOE Office of Science by Argonne National Laboratory under Contract No. DE-AC02-06CH11357. Funding for the synthesis and biochemical studies was provided by a "BIG" Award from the University of Chicago, the University of Chicago Women's Board, the National Institutes of Health (TM GM08720, Predoctoral Training Program in Chemistry and Biology, graduate fellowship to C.A.T.), the National Institute of General Medical Sciences (R35 GM119840 to B.C.D.), and start-up funds from the University of Chicago (S.A.S.).

## Author contributions
A.J. initiated the project, M.E. and R.J. cloned and expressed wild-type and mutant proteins, R.J. purified the first batch of protein and obtained the first wild-type PLpro crystal. C.T. purified proteins and crystallized proteins and complexes, while J.O. collected diffraction data, determined, refined and analyzed structures. Y.K. contributed to structure refinement and analysis of structural data. V.G.L., S.L.M., S.A.S., C.A.T., Y.Z., and Z.Z. designed and synthesized compounds **1–7**. S.A.A. and R.S.K. synthesized CV-2. K.J. performed all in vitro biochemical assays. G.R., V.N., and S.D. performed virus assay and analyzed data. Finally, A.J., B.C.D., and S.A.S. conceived of and directed the research as well as wrote the manuscript, while K.M. analyzed structural data and also wrote portions of the manuscript.

## Competing interests
The authors declare no competing interests.
