## [Peer Review File · Nature Communications]

REVIEWER COMMENTS

Reviewer #1 (Remarks to the Author):

1. the paper claims that they identified some crystal structures of SARS-CoV2 PLpro and some compounds would inhibit the activity of PLpro. It shows PLpro could be a promising target for COVID-19. This work is novel and would be of some interest to the community and the wider field. But they should provide more data to be convincing and strengthen the conclusions. (1) Could CoV2 PLpro cleave nsp1-nsp2 or nsp2-nsp3 or nsp4-nsp5? There should be a kind of protein cleavage experiments. LKGG-AmC is just a kind of peptide substrate. (2) For SARS-CoV PLPro, the consensus sequence is LRGG, but CoV2 PLpro is LKGG, there should be a reason. Since they do sequence alignments and structural comparison, they should give a reasonable explanation. (3) For CoV2 PLpro, besides C-H-D triad, there are some other residues essential for the enzymatic activity, such as oxygenanion hole binding residues. So it is better to produce some mutants and test the mutants's enzymatic activity to make the conclusions convincing. (4) As known for SARS-CoV and MERS-CoV PLpro having DeUB and DeISG activity, it is suggestive to investigate whether CoV2 PLpro has such kind of activity. (5) Since CoV-2 PLpro shows very high similarity to SARS-CoV PLpro, there should be some similarity and difference for them to bind the identified inhibitors.

2. There are also some advices for this work.

(1) There are papain-like protease and PLpro in the key words part. PLpro is abbreviation of papain-like protease. There is no need to put it twice.

(2) For the 8 crystal structures, two of them were overrefined. In the 2.70Å structure, the gap between R and Rfree is too big (30.88/23.51). And in the 2.09Å structure, the gap between R and Rfree is also too big (18.58/29.98).

(3) For the Supplementary figure1, it is better also to cite the paper about the crystal structure instead of only the PDB code.

(4) The kinetic parameters for CoV2 PLpro should be investigated, such as K_m and K_{cat} .

Reviewer #2 (Remarks to the Author):

In response to the SARS-CoV-2 pandemic, Osipiuk et al have investigated inhibition of the papain-like protease (PLpro) as a potential antiviral target by solving and assessing complexes with small molecules. Based on the previously identified SARS-CoV inhibitor GRL0617 (compound 1), the authors designed and synthesized a series of non-covalent inhibitors and report IC_{50} s as low as 2 μM and antiviral EC_{50} s as low as 1.5 μM . In addition to the inhibition data, the authors provide novel structures of WT PLpro and a C111S mutant; both variants are solved Apo and with an inhibitor bound. All the inhibitors bind in the P3/P4 binding site. Overall, this work was well written and would be valuable to the scientific community in our quest for antivirals against SARS-CoV2, after addressing the following concerns:

Major:

1. In Figure 1 the authors only show a one panel ribbon diagram of PLpro. To support the description in the text "The CoV-2 PLpro structure has "thumb-palm-fingers" architecture described before (Fig. 1). This arrangement is similar to the ubiquitin specific proteases (USPs), one of the five distinct deubiquitinating enzyme (DUB) families, despite low sequence identities (~10%)." This point would be much more convincing with a side-by-side comparison of these structures. Also if these overall structural features are going to be discussed they should be shown graphically perhaps by overlapping the catalytic domain in comparison with the SARS and MERS PLpro – an overall RMSD is not very informative.

2. What is the catalytic activity of the PLpro-WT and PLpro-C111S activities on LKGG-AMC with 1 μM enzyme and 40 μM LKGG-AMC substrate? This is mentioned in the methods. Was there any fluorescent interference with the inhibitors?

3. The antiviral data needs further elaboration, the controls need to be explicitly described, while

Figure 3 looks promising. What was the MOI of infection, what was the antiviral control used (NHC / Remdesivir)? Was there a DMSO control included for comparison? What was the cellular toxicity compared to a control? The errors should be given for the EC₅₀'s (Figure 3A). Unfortunately none of the novel inhibitors were better than the parent GRL0617.

4. Figure 4. Panel E should be in the same orientation as the other figures to best compare how these inhibitors fit in the P3 / P4 substrate binding sites. In panels A-D the residues coordinating the inhibitor that are conserved between the sequences shown in the alignment should be highlighted in some manner (either coloring them differently or their labels).

Minor Points

Page 6 – when talking about crystal packing the author's should explicitly refer to the space groups – not vaguely to differential packing.

Regarding the Xray Stats / Supplemental Table 1, most of the data looks reasonable. The only outlier would be 7JIR which had large R_{work}/R_{free} gap (18.6/30.0) for a 2.09 Å structure. Also, this table should probably have a row for number of molecules in the AU, which would make it easier to relate the number of ions per structure. Lastly, besides the structural Zinc, how do you decide which electron density should be a Zn or Cl ion as opposed to an H₂O?

Typo: At the top of page 4, did the authors mean to leave a doi hyperlink or should that be a typical reference citation?

Reviewer #3 (Remarks to the Author):

Osipiuk et al. describe GRL0617 and its derivatives, which target the papain-like protease (PLpro) of SARS-CoV-2. These compounds can inhibit the viral protease in vitro and also demonstrate antiviral activity at micromolar level. Further the authors report the crystal structures of wild-type PLpro, the active site C111S mutant, and their complexes, describing the binding modes of GRL0617 and its derivatives. Compared with other two classical drug targets (Mpro and RdRP), PLpro is also gradually attracting attention due to its significant roles both in replicase polyprotein cleavage and in antagonizing immune response. GRL0617 has been reported as an inhibitor against SARS-CoV PLpro. So it is not surprising this compound and its derivatives also target SARS-CoV-2 PLpro. It is a pity that these derivatives did not show improved activity compared with GRL0617. Nonetheless, given the urgency to find effective remedies against COVID-19, the findings reported in this paper are still useful to others in the field. But there still remain some major concerns to be addressed.

1. GRL0617 and its derivatives inhibit SARS-CoV-2 PLpro differently. The authors need to address the reasons account for the differences from structural analysis.
2. Please show the compounds with the omit maps in the supplementary materials.
3. In supplemental Table 1, R_{free} and R_{work} have a huge gap both for PLpro WT/100K/2.70 Å and PLpro C111S mutant-1/100K/2.09 Å. These two structures need to be further refined.
4. In the methods/PLpro inhibition assay, "Reactions containing varying concentrations of inhibitor (0-500 μM) and PLpro enzyme (0.3 μM) in Tris-HCl pH 7.3, 1 mM EDTA were incubated for approximately five minutes". It is weird that authors added EDTA to test enzyme activity/inhibition since PLpro does have a zinc finger motif. The authors need to repeat the experiments with and without EDTA.

The haste in which this manuscript has been prepared may have contributed to a number of sloppy mistakes and the composition errors that I noticed are listed below.

1. "Severe Acute Respiratory Syndrome Coronavirus 2" should be "severe acute respiratory syndrome coronavirus 2"
2. "genus Betacoronavirus (should be in italic type)".
3. "In CoV-2" should be "In SARS-CoV-2".
4. "SARS-CoV-1" should be "SARS-CoV".

Structure of papain-like protease from SARS-CoV-2 and its complexes with non-covalent inhibitors, revision 1.

The detailed response to the reviewer comments and questions is provided below.

Response to Reviewer #1

1. the paper claims that they identified some crystal structures of SARS-CoV2 PLpro and some compounds would inhibit the activity of PLpro. It shows PLpro could be a promising target for COVID-19. This work is novel and would be of some interest to the community and the wider field. But they should provide more data to be convincing and strengthen the conclusions.

(1) Could CoV2 PLpro cleave nsp1-nsp2 or nsp2-nsp3 or nsp4-nsp5? There should be a kind of protein cleavage experiments. LKGG-AmC is just a kind of peptide substrate.

Response: In SARS-CoV-2, there are three sites conserved in thousands of SARS-CoV-2 gRNA sequences. There is only one copy of PLpro in this genome and the enzyme cleaves these three sites between Nsp1 and Nsp2 (LN \downarrow GGAYTR), Nsp2 and Nsp3 (LK \downarrow GGAPT \downarrow K), and Nsp3 and Nsp4 (LK \downarrow GGKIVN) liberating Nsp1, Nsp2 and Nsp3. For our assay development we used three sites (LN \downarrow GG, LK \downarrow GG and LR \downarrow GG) and we found that LKGG was cut the most efficiently. We have also found that Leu residue is essential for peptide recognition. Proteolytic activity of recombinant PLpro from SARS-CoV was extensively characterized in in vitro and in vivo assays (for example Harcourt et al. *J. Virol.* 2004 and other references). Also, one of the inhibitors described in our paper binds to the same site in SARS-CoV PLpro. These proteins are very similar and have identical active sites, we expect SARS-CoV-2 behave exactly this same. Recent publications confirmed that recombinant SARS-CoV-2 PLpro can recognize the LRGG sequence in the context of larger polypeptides. It was shown that PLpro is active against ubiquitin and ISG15 substrates (Shin, D. et al. *Nature* **78**, 13600–6 (2020), Klemm, T. et al. *EMBO J* **39**, e106275 (2020)). We believe, repeating these experiments would be redundant.

(2) For SARS-CoV PLpro, the consensus sequence is LRGG, but CoV2 PLpro is LKGG, there should be a reason. Since they do sequence alignments and structural comparison, they should give a reasonable explanation.

Response: As discussed in response to question (1) above, SARS-CoV-2 PLpro recognizes three sites in Pp1a and Pp1ab (LN \downarrow GG and two LK \downarrow GG). In addition, it also recognizes conserved LRGG sites found in C-termini of ubiquitin and ISG15 proteins (Shin, D. et al. *Nature* **78**, 13600–6 (2020), Klemm, T. et al. *EMBO J* **39**, e106275 (2020)). Our structures showed specific accommodation of Leu side chain in P4 site, hydrophilic side chain in P3 site and only glycine residues can be accepted into P2 and P1 sites.

(3) For CoV2 PLpro, besides C-H-D triad, there are some other residues essential for the enzymatic activity, such as oxyanion hole binding residues. So it is better to produce some mutants and test the mutants's enzymatic activity to make the conclusions convincing.

Response: We have made four mutations in this PLpro (C111S, C155S, C181A, C248R). We have only characterized C111S mutation, which makes enzyme inactive (Supplemental Fig. 3). The mechanism of cysteine proteases have been studied extensively. However, the role of an oxyanion hole in PLpro is still somewhat controversial. Previous studies predicted that the oxyanion is within hydrogen-bonding distance of Cys111 with the Trp side chain (Trp106) (Ratia et al., 2006) but in our structure it seems too far (4.6 Å). High resolution studies of the PLpro

complex with ubiquitin suggested that the oxyanion hole is provided by the main chain amides (Chou et al. 2014). We included the following text: “In agreement with previous suggestions our structures point to a potential oxyanion hole provided by main chain amides of Cys111/Tyr112.” We believe that detailed mechanistic investigation of the PLpro is beyond the scope of this manuscript as the characterized inhibitors do not bind directly to the active site.

(4) As known for SARS-CoV and MERS-CoV PLpro having DeUB and DeISG activity, it is suggestive to investigate whether CoV2 PLpro has such kind of activity.

Response: Yes, this is indeed an important point. It was reported recently in two manuscripts by Shin, D. *et al. Nature* **78**, 13600–6 (2020) and Klemm, T. *et al. EMBO J* **39**, e106275 (2020) that PLpro shows such activity.

(5) Since CoV-2 PLpro shows very high similarity to SARS-CoV PLpro, there should be some similarity and difference for them to bind the identified inhibitors.

Response: The PLpros from SARS-CoV and SARS-CoV-2 are very similar and therefore expected to bind inhibitors similarly. In fact, we have shown that SARS-CoV PLpro and SARS-CoV-2 PLpro bind compound 1 in this same location and with very comparable affinity. In contrast, PLpro from MERS is very different and is not inhibited by compound 1. We have indicated in Fig. 4 conservation of residues in the inhibitor binding site. It is possible that other pockets of PLpro that are less conserved can be exploited for CoV-2 specific inhibitors.

2. There are also some advices for this work.

(1) There are papain-like protease and PLpro in the key words part. PLpro is abbreviation of papain-like protease. There is no need to put it twice.

Response: We deleted PLpro in abbreviations.

(2) For the 8 crystal structures, two of them were overrefined. In the 2.70Å structure, the gap between R and Rfree is too big (30.88/23.51). And in the 2.09Å structure, the gap between R and Rfree is also too big (18.58/29.98).

Response: We decided to remove the 2.7 Å structure of PLpro

We decided to remove the original structure of the PLpro protein at 2.7 Å resolution from the manuscript, because 1.79 Å high resolution apo structure provides all key information. The 2.7 Å structure was very important because it was the very first structure of the PLpro available to biology community. However, due to reviewers concern about structure quality and due to superiority of the structure at 1.79 Å resolution which was deposited later, we decided to remove it from the manuscript. We believe that reason for big gap between R-work and R-free factors is poor quality of the crystal. Nevertheless, we decided to improve the 2.7 Å structure by applying non-crystallographic symmetry refinement and following the 1.79 Å structure in protein modelling in questionable areas. The improved structure was submitted to PDB to replace the existing structure. The new R-work and R-free factors are equal to 23.5 and 27.9 %, respectively. All discussions in revised manuscript are based on the high resolution apo structure. For 2.09 Å (mutant compound 1) structure there was a typo for Rfree value in supplementary Table 1, the Rfree value should be 19.98 (not 29.98) as shown in PDB deposit 7JIR.

(3) For the Supplementary figure1, it is better also to cite the paper about the crystal structure instead of only the PDB code.

Response: We cite in the Supplementary Fig. 1 all references for PLpro PDB deposits.

(4) The kinetic parameters for CoV2 PLpro should be investigated, such as K_m and K_{cat} .

Response: We believe that investigation of kinetic parameters is beyond scope of this 61 pages long manuscript.

Response to Reviewer #2:

In response to the SARS-CoV-2 pandemic, Osipiuk et al have investigated inhibition of the papain-like protease (PLpro) as a potential antiviral target by solving and assessing complexes with small molecules. Based on the previously identified SARS-CoV inhibitor GRL0617 (compound 1), the authors designed and synthesized a series of non-covalent inhibitors and report IC_{50} s as low as 2 μM and antiviral EC_{50} s as low as 1.5 μM . In addition to the inhibition data, the authors provide novel structures of WT PLpro and a C111S mutant; both variants are solved Apo and with an inhibitor bound. All the inhibitors bind in the P3/P4 binding site. Overall, this work was well written and would be valuable to the scientific community in our quest for antivirals against SARS-CoV2, after addressing the following concerns:

Major:

1. In Figure 1 the authors only show a one panel ribbon diagram of PLpro. To support the description in the text “The CoV-2 PLpro structure has “thumb–palm–fingers” architecture described before (Fig. 1). This arrangement is similar to the ubiquitin specific proteases (USPs), one of the five distinct deubiquitinating enzyme (DUB) families, despite low sequence identities (~10%).” This point would be much more convincing with a side-by-side comparison of these structures. Also if these overall structural features are going to be discussed they should be shown graphically perhaps by overlapping the catalytic domain in comparison with the SARS and MERS PLpro – an overall RMSD is not very informative.

Response: As requested by reviewer we added to Fig. 1 structure of human USP12 which has similar fold and architecture of catalytic domains. We have also added supplemental Fig. 5 that compares PLpro catalytic domains of SARS-CoV-2 and MERS-CoV.

2. What is the catalytic activity of the PLpro-WT and PLpro-C111S activities on LKGG-AMC with 1 μM enzyme and 40 μM LKGG-AMC substrate? This is mentioned in the methods. Was there any fluorescent interference with the inhibitors?

Response: Supplemental Fig. 3 shows that there is no enzyme activity observed for C111S mutant with 1 μM enzyme and 40 μM LKGG-AMC substrate. We have tested now over hundred different compounds and very few showed interference with the fluorescence probe. No interference was observed for 7 compounds reported in this manuscript.

3. The antiviral data needs further elaboration, the controls need to be explicitly described, while Figure 3 looks promising. What was the MOI of infection, what was the antiviral control used (NHC / Remdesivir)? Was there a DMSO control included for comparison? What was the

cellular toxicity compared to a control? The errors should be given for the EC50's (Figure 3A). Unfortunately none of the novel inhibitors were better than the parent GRL0617.

Response: We have added the requested information to Fig. 3 (EC50 error) and the material and methods. Briefly, Vero E6 cells were infected at at MOI of 0.1 for 24 hours. The positive antiviral control was the parent molecule (GRL0617) and all data was the 0 drug concentration was DMSO control. No effects of cell viability was observed under the concentrations tested as reflected by total cell number by microscopy.

4. Figure 4. Panel E should be in the same orientation as the other figures to best compare how these inhibitors fit in the P3 / P4 substrate binding sites. In panels A-D the residues coordinating the inhibitor that are conserved between the sequences shown in the alignment should be highlighted in some manner (either coloring them differently or their labels).

Response: We have edited Figure 4 to emphasize sequence conservation. Labels for conserved residues are in red. However, in order to visualize the narrow tunnel leading to the active and P1-P4 sites we have to adjust the orientation.

Minor Points

Page 6 – when talking about crystal packing the author's should explicitly refer to the space groups – not vaguely to differential packing.

Response: Our structures were determined in three different space groups (C2, P3₂21, I4₁22) showing that crystal packing has very little impact on the structure and ligand binding. We commented in the text.

Regarding the Xray Stats / Supplemental Table 1, most of the data looks reasonable. The only outlier would be 7JIR which had large Rwork/Rfree gap (18.6/30.0) for a 2.09 Å structure. Also, this table should probably have a row for number of molecules in the AU, which would make it easier to relate the number of ions per structure. Lastly, besides the structural Zinc, how do you decide which electron density should be a Zn or Cl ion as opposed to an H2O?

Response: We responded to reviewer one about Rfree typo, the correct number is 19.98 in place of 29.98 as shown in PDB deposit 7JIR. In the S Table 1 we included a row with number of molecules in the AU. Here is explanation about assignment of ions in the electron density. For zinc, the determination was based mainly on anomalous maps showing the positions of heavy atoms in the structures. Diffraction data was collected at 12.66 keV energy which is above absorption-edge energy for zinc. This allowed us to create anomalous maps showing zinc atoms at the map contours equal to or higher than 9 sigma. In addition, all zinc ions in the structures are coordinated to negatively-charged side-chains of cysteines, histidines, aspartates and/or glutamates. Zinc was present in bacteria cell growth media and supplemented during protein purification. Decision to place chloride ions in the structures was based mainly on mFo-DFc difference maps in places where, after placement of water molecules, additional peaks in electron density map were still clearly visible. These sites showed electron densities higher than 4.5 sigma on 2mFo-DFc maps (with exception of chloride ions at partial occupancies). All chloride ions are coordinated to positively charged atoms in the structures. These sites do not show significant electron densities in anomalous maps.

Typo: At the top of page 4, did the authors mean to leave a doi hyperlink or should that be a typical reference citation?

Response: We included two resent references for Nsp1 (PMID: 32908316, PMID: 32680882)

Reviewer #3 (Remarks to the Author):

Osipiuk et al. describe GRL0617 and its derivatives, which target the papain-like protease (PLpro) of SARS-CoV-2. These compounds can inhibit the viral protease in vitro and also demonstrate antiviral activity at micromolar level. Further the authors report the crystal structures of wild-type PLpro, the active site C111S mutant, and their complexes, describing the binding modes of GRL0617 and its derivatives. Compared with other two classical drug targets (Mpro and RdRP), PLpro is also gradually attracting attention due to its significant roles both in replicase polyprotein cleavage and in antagonizing immune response. GRL0617 has been reported as an inhibitor against SARS-CoV PLpro. So it is not surprising this compound and its derivatives also target SARS-CoV-2 PLpro. It is a pity that these derivatives did not show improved activity compared with GRL0617. Nonetheless, given the urgency to find effective remedies against COVID-19, the findings reported in this paper are still useful to others in the field. But there still remain some major concerns to be addressed.

1. GRL0617 and its derivatives inhibit SARS-CoV-2 PLpro differently. The authors need to address the reasons account for the differences from structural analysis.

Response: GRL0617 and its derivatives appear to bind to the same site (as we shown for compounds 2 and 3) but showed different affinity, as we discussed in the text. However, these compounds have different cellular efficacy. This cannot be addressed without significant characterization of other derivatives.

2. Please show the compounds with the omit maps in the supplementary materials.

Response: As suggested by reviewer we included omit maps in the supplemental Fig. X.

3. In supplemental Table 1, Rfree and Rwork have a huge gap both for PLpro WT/100K/2.70 Å and PLpro C111S mutant-1/100K/2.09 Å. These two structures need to be further refined.

Response: Please see response to reviewer 1.

4. In the methods/PLpro inhibition assay, “Reactions containing varying concentrations of inhibitor (0-500 μM) and PLpro enzyme (0.3 μM) in Tris-HCl pH 7.3, 1 mM EDTA were incubated for approximately five minutes”. It is weird that authors added EDTA to test enzyme activity/inhibition since PLpro does have a zinc finger motif. The authors need to repeat the experiments with and without EDTA.

Response: PLpro is not a metallo enzyme, although it has a tightly bound structural zinc ion. We repeated the probe characterization in the presence and absence of EDTA and see no significant differences in activity (new data in supplemental Figure 3D). The conditions for assay were taken from published data.

The haste in which this manuscript has been prepared may have contributed to a number of sloppy mistakes and the composition errors that I noticed are listed below.

1. “Severe Acute Respiratory Syndrome Coronavirus 2” should be “severe acute respiratory syndrome coronavirus 2”

Response: We capitalize first letters to emphasize SARS CoV-2 abbreviation, we underlined these letters.

2. "genus *Betacoronavirus* (should be in italic type)".

Response: Corrected.

3. "In CoV-2" should be "In SARS-CoV-2".

Response: Corrected.

4. "SARS-CoV-1" should be "SARS-CoV".

Response: Corrected.

REVIEWERS' COMMENTS

Reviewer #1 (Remarks to the Author):

The resubmitted manuscript made great improvements and the results presented would lay the foundation for the discovery and optimization of drugs against COVID-19. I suggested it could be accepted for publication in NC. And I have some advices:

- (1) In figure 1A, it should be Ubl instead of Ub, because in PLpro, it has four domains including ubiquitin-like domain (Ubl).
- (2) In line 124, It is SADS PLP2 instead of "SADS PLpro" because in SADS-CoV, nsp3 contains two PLPro, PLP1 and PLP2. Up to now, only the crystal structure of SADS-CoV PLP2 has been determined and reported. It should be clearly described. Otherwise, there will cause confusions. All coronaviruses, no matter of which genus, have a papain-like protease domain contained within nsp3. What is special for alphacoronaviruses and betacoronaviruses of clade a is that they have both PLP1 and PLP2.

Reviewer #2 (Remarks to the Author):

The updated manuscript with the revised figures and error analysis really highlight the significance of this important manuscript describing the structure and inhibition of the PLpro enzyme of SARS-CoV-2